# Numerical Analysis of Thermal Radiative Maxwell Nanofluid Flow Over-Stretching Porous Rotating Disk

**DOI:** 10.3390/mi12050540

**Published:** 2021-05-10

**Authors:** Shuang-Shuang Zhou, Muhammad Bilal, Muhammad Altaf Khan, Taseer Muhammad

**Affiliations:** 1School of Science, Hunan City University, Yiyang 413000, China; zhoushuangshuang@hncu.edu.cn; 2Department of Mathematics, City University of Science and Information Technology, Peshawar 25000, Pakistan; 3Institute for Groundwater Studies, Faculty of Natural and Agricultural Sciences, University of Free State, Bloemfontein 9300, South Africa; altafdir@gmail.com; 4Department of Mathematics, College of Sciences, King Khalid University, Abha 61413, Saudi Arabia; taseer_qau@yahoo.com; 5Mathematical Modelling and Applied Computation Research Group (MMAC), Department of Mathematics, King Abdulaziz University, P.O. Box 80203, Jeddah 21589, Saudi Arabia

**Keywords:** bvp4c, RK4 technique, brownian motion, porous rotating disk, maxwell nanofluid, thermally radiative fluid, von karman transformation

## Abstract

The fluid flow over a rotating disk is critically important due to its application in a broad spectrum of industries and engineering and scientific fields. In this article, the traditional swirling flow of Von Karman is optimized for Maxwell fluid over a porous spinning disc with a consistent suction/injection effect. Buongiorno’s model, which incorporates the effect of both thermophoresis and Brownian motion, describes the Maxwell nanofluid nature. The dimensionless system of ordinary differential equations (ODEs) has been diminished from the system of modeled equations through a proper transformation framework. Which is numerically computed with the bvp4c method and for validity purposes, the results are compared with the RK4 technique. The effect of mathematical abstractions on velocity, energy, concentration, and magnetic power is sketched and debated. It is perceived that the mass transmission significantly rises with the thermophoresis parameter, while the velocities in angular and radial directions are reducing with enlarging of the viscosity parameter. Further, the influences of thermal radiation Rd and Brownian motion parameters are particularly more valuable to enhance fluid temperature. The fluid velocity is reduced by the action of suction effects. The suction effect grips the fluid particles towards the pores of the disk, which causes the momentum boundary layer reduction.

## 1. Introduction

The researchers have been interested in Maxwell nanofluid flow over a porous spinning disc because of its many uses in engineering and innovation. Non-Newtonian fluids are important in a variety of manufactured liquids, including plastics, polymers, pulps, toothpaste and fossil fluids. To simulate the analysis of these liquids, a variety of models have been suggested. Shear stress and shear rate are linked in non-Newtonian liquids because of their nonlinear existence. The momentum equation in these fluids involves dynamic nonlinear terms, making it difficult to solve. A variety of mathematical models exist in the literature to simulate the performance of these fluids.

In the present era, the role of nanotechnology to fulfill the increasing demand for energy and face energy challenges is remarkable. The usage of nanoparticles in ordinary base fluid (water, kerosene oil, etc.) effectively enhances the heat transfer and improves their thermal properties. The applications of nanoparticles in certain fields of engineering and industry are in the cooling systems of electronic devices and in cancer therapy, heat exchangers, transformer cooling, nuclear reactors, and space cooling systems. Due to the ability of oil wetting and dispersing, they are also used for cleaning purposes, in power generation, microfabrication, hyperthermia, and metallurgical purposes.

## 2. Literature Review

The rotating disk phenomena are widely used in centrifugal filtration, turbomachines, the braking system of vehicles, jet motors, sewing machines, turbine systems, heat exchangers and computer disk drives, etc. Von Karman [1] for the first time introduced similarity transformation. To solve Navier-Stokes equations, he studied fluid flow over an infinite rotating disk. Cochran [2] employed Von Karman’s similarity transformation to incompressible fluid over a rotating disk and examined the asymptotic solution. Wagner [3] investigated the mechanism of heat transfer over the rotating disk, by considering Von Karman’s velocity distribution, and analyzed convection in the non-turbulent flow. Turkyimazogl [4] looked at fluid movement over a spinning disc that was stretching under the influence of a static electric field. Liang et al. [5] reported a comparative study between semi-analytical model and experimental data to yield the best settlement. Millsaps and Pohlhausen [6] used Von Karman’s similarity approach and analyzed heat distribution with the consequences of entropy generation over revolving disk. The three-dimensional (3D) magnetohydrodynamics (MHD) stagnation flow of ferrofluid, the numerical solution was revealed by Mustafa et al. [7]. Mustafa et al. [8], by taking MHD nanofluid over rotating surface with the effects of partial slips, observed that the boundary layer thickness and momentum transport are reduced due to slip effects. Rashidi et al. [9] have used a spinning disc to perform a viscous dissipation review for MHD nanofluid.

The people of the modern world are facing many challenges due to the increasing demand for energy by the latest technologies. Firstly Choi [10] presented the nanofluids terminology. The Brownian motion and thermophoresis mechanisms bring about a significant role in improving the thermal properties of base fluid presented by Buongiorno [11]. Turkyilmazolglu [12] analytically studied the energy and momentum equations of nanofluid flow, to deduce heat and flow transport. Pourmehran [13] considered Cu and Al_2_O_3_ nanoparticles to study heat and flow transfer in the microchannel. Hatami et al. [14] reported the heat transfer in nanofluid with the phenomena of natural convection. The Oldroyd-B fluid with nanoparticles over stretching sheet surface was reported by Nadeem et al. [15]. Aziz and Afify [16] have used the technique of the Lie group, to study non-Newtonian nanofluids. Yang et al. [17] studied the convective heat with Buongiorno Model’s for nanofluid in the concentric annulus.

The study of a Newtonian fluid, due to its wide applicability in different fields of science and engineering, attracted the attention of scientists and researchers during the last few decades. Its major role is in geophysics, polymer solution, paper production, cosmetic processes, exotic lubricants, paints, suspensions, colloidal solutions, nuclear and chemical industries, pharmaceuticals, oil reservoirs, bioengineering, etc. [18]. Xiao et al. [19] attempted a fractal model for the capillary flow through a torturous capillary with the non-smooth surface in porous media. Attia [20] evaluated the Reiner-Rivlin numerical simulations for thermal convection over a porous spinning disc qualitatively. Griffiths [21] tested the Newtonian fluid Carreau viscosity model and high shear stresses on spinning discs. The numerical analysis of Reiner-Rivlin fluid flow for heat transfer and slip flow over a spinning disc is treated by Mustafa and Tabassum [22]. The micropolar fluid for thermophoretic diffusion generated by the rotation of the disk was examined by Doh and Muthtamilselvan [23].

Darcy’s law is a mathematical equation that explains how fluid flows through a porous medium. Henry Darcy developed the law based on the effects of studies on the flow of water across sand beds, laying the groundwork for hydrogeology, a branch of earth sciences [24]. Fourier’s law in heat conduction, Ohm’s law in electrical networks, and Fick’s law in diffusion theory are all examples of this law. Morris Muskat [25] improved Darcy’s equation for a single-phase flow by incorporating viscosity into Darcy’s single (fluid) phase equation. It is easy to see that viscous fluids have a harder time passing through a porous medium than less viscous fluids. Rasool et al. [26,27,28] numerically simulated the Darcy-Forchheimer effect on MHD nanoliquid flow between stretching non-linear sheets. Rasool et al. [29] scrutinized the consequences of thermal radiation, chemical reaction and Dufour-Soret on incompressible steady Darcy-Forchheimer flow of nanoliquid. Shafiq et al. [30] studied nanofluid flow under the influence of convective boundary conditions and thermal slip over a spinning frame. They found that the axial and transverse velocity fields all drop significantly due to the Forchheimer number’s strong retardation. Skin friction is intensified by the Forchheimer number and porosity ratios, while skin friction is diminished by all slip parameters.

Viscoelastic fluids are a subclass of Newtonian fluid having memory effects. The intensity of energy discharged by these fluids is mainly accountable for recovery after the stress is removed. The Maxwell flow regime is the most basic viscoelastic fluid model, expressing memory effects by fluid relaxation time [31]. The attitude of the current model is very close to that of other geomaterials and polymers models. The aim of the present work is to provide a mathematical model for unsteady boundary layer flow of non-Newtonian Maxwell nanofluid with the heat transmission over a porous spinning disc. The present work has many industrial and engineering applications, which increases its worth. Using a resemblance method, the system of ODEs is limited to a structure of PDEs. A boundary value solver (bvp4c) technique is used to draw a numerical solution to the problem while RK4 method has been applied for validity.

## 3. Formulation of the Problem

Consider an unsteady hybrid nanoliquid flow over a stretching porous spinning disc. The magnetic force $B_{0}$ is introduced to the disc vertically. The disc rotates and stretches at different speeds (u,v)=(cr,cΩ), where *c* and *w* are the spinning and extending rates, respectively. The disk temperature is represented by $\tau_{w}$. The formulation of the problems is conducted in (r,φ,z) cylindrical coordinates, where u,v,w is velocity component increasing in (r,φ,z) direction. At z− the axis, the motion of the disk is assumed to be axisymmetric. The thermal radiation is significant in modeling the energy equation. The viscosity of a fluid is taken to be temperature-dependent $\mu \left(\tau \right)=\mu_{0}e^{-\zeta \left(\tau -\tau_{0}\right)}$. The concentration and temperature are represented by $\left(C_{w},\tau_{w}\right)$ and $\left(C_{\infty },\tau_{\infty }\right)$ represent the concentration and temperature above the disk surface.

### 3.1. Governing Equations

Under the presuppositions stated above, the flow equations are as observes [31]:(1)∇.V=0,
(2)$$\rho_{f}\left(V.\nabla \right)V=\nabla p+\nabla .S+j\times B,$$
(3)$$\left(V.\nabla \right)\tau =\alpha \nabla^{2}\tau +\tau^{*}\left(D_{B}\nabla C.\nabla \tau +\frac{D\tau }{\tau_{\infty }}\nabla \tau .\nabla \tau \right)+\nabla .q_{rad,}$$
(4)$$\left(V.\nabla \right)C=D_{B}\nabla^{2}C+\frac{D\tau }{\tau_{\infty }}\nabla^{2}\tau ,$$
(5)∇.B=0,
(6)$$\rho \frac{\partial B}{\partial t}=\rho \nabla +\left(V\times B\right)+\frac{\rho }{\sigma \mu_{2}}\nabla^{2}V.$$
where $D_{B},D_{T},\;\rho_{f},V$ and α are the coefficients of Brownian motion, thermophoretic diffusion, fluid density, velocity, and thermal and diffusivity respectively. Equations (1)–(6) are simplified because of the boundary layer approximation concept: (7)$$\frac{\partial u}{\partial r}+\frac{u}{r}+\frac{\partial w}{\partial z}=0,$$
(8)$$\frac{\partial u}{\partial t}+u\frac{\partial u}{\partial r}+w\frac{\partial u}{\partial z}-\frac{v^{2}}{r}=\left[\begin{array}{l} \frac{1}{\rho }\left(\mu \left(\tau \right)\frac{\partial u}{\partial z}\right)-\lambda_{1}(u^{2}\frac{\partial^{2}u}{\partial r^{2}}+w\frac{\partial^{2}u}{\partial z^{2}}+2uw\frac{\partial^{2}u}{\partial r\partial z}-\frac{2uv}{r}\\ \frac{\partial v}{\partial r}-\frac{2vw}{r}\frac{\partial v}{\partial z}+\frac{uv^{2}}{r^{2}}+\frac{v^{2}}{r^{2}}\frac{\partial u}{\partial r})-\frac{\sigma B_{0}^{2}}{\rho }\left(u+w\lambda_{1}\frac{\partial u}{\partial z}\right) \end{array}\right],$$
(9)$$\frac{\partial v}{\partial t}+u\frac{\partial v}{\partial r}+w\frac{\partial v}{\partial z}-\frac{uv}{r}=\left[\begin{array}{l} \frac{1}{\rho }\left(\mu \left(\tau \right)\frac{\partial v}{\partial z}\right)-\lambda_{1}(u^{2}\frac{\partial^{2}v}{\partial r^{2}}+w^{2}\frac{\partial^{2}v}{\partial z^{2}}+2uw\frac{\partial^{2}v}{\partial r\partial z}+\frac{2uv}{r}\\ \frac{\partial u}{\partial r}+\frac{2vw}{r}\frac{\partial u}{\partial z}+\frac{2u^{2}v}{r^{2}}-\frac{v^{2}}{r^{2}}\frac{\partial v}{\partial r})-\frac{\sigma B_{0}^{2}}{\rho }\left(v+w\lambda_{1}\frac{\partial v}{\partial z}\right) \end{array}\right],$$
(10)$$\frac{\partial \tau }{\partial t}+u\frac{\partial \tau }{\partial r}+w\frac{\partial \tau }{\partial z}=\frac{k}{\rho_{c_{p}}}\left(\frac{\partial^{2}\tau }{\partial z^{2}}\right)+\tau^{*}(D_{B}\frac{\partial \tau }{\partial z}\frac{\partial C}{\partial z}+\frac{D_{\tau }}{\tau_{\infty }}\left(\frac{\partial \tau }{\partial z}{)}^{2}\right)-\frac{1}{\rho_{c_{p}}}\frac{\partial q_{r}}{\partial z},$$
(11)$$\frac{\partial C}{\partial t}+u\frac{\partial C}{\partial r}+w\frac{\partial C}{\partial z}=D_{B}\left(\frac{\partial^{2}C}{\partial z^{2}}\right)+\frac{D_{\tau }}{\tau_{\infty }}\left(\frac{\partial^{2}\tau }{\partial z^{2}}\right),$$
(12)$$\frac{\partial B_{r}}{\partial t}=\left[-w\frac{\partial B_{r}}{\partial z}-B_{r}\frac{\partial w}{\partial z}+u\frac{\partial B_{z}}{\partial z}+B_{z}\frac{\partial u}{\partial z}+\frac{1}{\sigma \mu_{2}}\left(\frac{\partial^{2}B_{r}}{\partial r^{2}}+\frac{\partial^{2}B_{r}}{\partial z^{2}}+\frac{1}{r}\frac{\partial B_{r}}{\partial r}-\frac{B_{r}}{r^{2}}\right)\right],$$
(13)$$\frac{\partial B_{z}}{\partial t}=\left[w\frac{\partial B_{r}}{\partial r}+B_{r}\frac{\partial w}{\partial r}+\frac{1}{r}wB_{r}-u\frac{\partial B_{z}}{\partial r}-B_{z}\frac{\partial u}{\partial r}-\frac{1}{r}uB_{z}+\frac{1}{\sigma \mu_{2}}\left(\frac{\partial^{2}B_{z}}{\partial r^{2}}+\frac{\partial^{2}B_{z}}{\partial z^{2}}+\frac{1}{r}\frac{\partial B_{z}}{\partial r}\right)\right].$$

The temperature difference within a flow is assumed to be small, therefore higher-order in Taylor series and ignored at $\tau_{\infty }$, by using Rosseland approximation, the simple form of radiation heat flux is as follow [31]: (14)$$q_{r}=\frac{-4\sigma^{*}\partial \tau^{4}}{3k^{*}\partial z}=-\frac{16\sigma^{*}\tau^{3}}{3k^{*}}\frac{\partial \tau }{\partial z},$$
using Equation (14) in (10), we get
(15)$$\frac{\partial \tau }{\partial t}+u\frac{\partial \tau }{\partial r}+w\frac{\partial \tau }{\partial z}=\frac{k}{\rho_{c_{p}}}\left(\frac{\partial^{2}\tau }{\partial z^{2}}\right)+\tau^{*}(D_{B}\frac{\partial \tau }{\partial z}\frac{\partial C}{\partial z}+\frac{D_{\tau }}{\tau_{\infty }}\left(\frac{\partial \tau }{\partial z}{)}^{2}\right)-\frac{16\sigma^{*}\tau^{3}}{3k^{*}}\frac{\partial \tau }{\partial z}.$$

The boundary conditions are: (16)$$\begin{array}{l} u=cr,v=\Omega r,\;w=W,\;C=C_{w},\;T=T_{w},\;B_{r}=1,B_{z}=1\;\;\;\;atz=0\\ u\to 0,\;v\to 0,\;w\to 0,\;C\to C_{\infty },\;T\to T_{\infty },\;B_{r}\to 0,B_{z}\to 0\;\;as\;\;\;\;Z\to \infty . \end{array}$$

### 3.2. Similarity Transformation

Considering the following transformation, to reduce the system of PDEs to the system of ODEs: (17)$$\begin{array}{l} u=\frac{cr}{1-\alpha t}F\left(\eta \right),\nu =\frac{\Omega r}{1-\alpha t}G\left(\eta \right),\;w=\frac{\sqrt{cv}}{1-\alpha t}H\left(\eta \right),\eta =\sqrt{\frac{c}{v}}z,\;B_{r}=\frac{crM_{0}}{1-\alpha t}M\prime \left(\eta \right),\\ B_{z}=\frac{-M_{0}{\left(2\nu_{f}c\right)}^{\frac{1}{2}}}{1-\alpha t}N\left(\eta \right),\;\;T=\left(T_{\infty }\right)+\Theta \left(\eta \right)\left(T_{W}-T_{\infty }\right)C=\left(C_{\infty }\right)+\phi \left(\eta \right)\left(C_{W}-C_{\infty }\right). \end{array}\}$$

The following system of ODEs is obtained by using Equation (17) in Equations (7)–(13) and (15) and (16): (18)$$F^{\prime \prime }=F^{\prime }\Theta^{\prime }+\frac{e^{\delta \Theta }}{\delta }\left\{S\left(\frac{F^{\prime }\eta }{2}+F\right)+F^{2}+HF^{\prime }-G^{2}\right\}+\beta_{1}\frac{e^{\delta \Theta }}{\delta }\left\{HF^{\prime }+2FF^{\prime }H-2HGG^{\prime }\right\}-M\frac{e^{\delta \Theta }}{\delta }\left(F+\beta_{1}HF^{\prime }\right),$$
(19)$$G^{\prime \prime }=\frac{\delta G^{\prime }\Theta^{\prime }+e^{\delta \Theta }\left\{S\left(\frac{G^{\prime }-\eta }{2}+G\right)+2FG+HG^{\prime }\right\}+\beta_{1}e^{\delta \Theta }\left\{2FHG^{\prime }+2FHG\right\}+Me^{\delta \Theta }\left(G+\beta_{1}HG^{\prime }\right)}{1+\beta_{1}H^{2}e^{\delta \Theta }},$$
(20)$$\Theta^{\prime \prime }=\frac{-4Rd\Theta^{2}{\Theta^{\prime }}^{2}{\left(\Theta -1\right)}^{3}-6{\Theta^{\prime }}^{2}\Theta {\left(Q_{w}-1\right)}^{2}-3{\Theta^{\prime }}^{2}\left(Q_{w}-1\right)-\mathrm{Pr}\left(-\frac{S}{2}\left(\Theta^{\prime }\eta \right)-H\Theta^{\prime }+Nb\Theta \phi^{\prime }+Nt{\Theta^{\prime }}^{3}\right)}{\left(1+\frac{4}{3}\right)Rd+\frac{4}{3}Rd\Theta^{3}{\left(Q_{w}-1\right)}^{3}+3\Theta^{2}{\left(Q_{w}-1\right)}^{2}+3\Theta \left(\Theta -1\right)},$$
(21)$$\phi^{\prime \prime }=Sc\left(A\phi^{\prime }\eta +H\phi^{\prime }\right)-\frac{Nt}{Nb}\Theta^{\prime \prime },$$
(22)$$M^{\prime \prime \prime }=Bt\left[-HM^{\prime \prime }+M^{\prime }H^{\prime }+FN^{\prime }+NF^{\prime }+S\left(\frac{M^{\prime \prime }\eta }{2}+M^{\prime }\right)\right],$$
(23)$$M^{\prime \prime }=-Bt\left[2HM^{\prime }+2NF-\frac{S}{2}\left(N\eta +N\right)\right].$$
the transforms conditions are: (24)$$\begin{array}{l} F\left(0\right)=1,\;G\left(0\right)=\omega ,\;H=W_{s},\;\Theta \left(0\right)=1,\;f\left(0\right)=1,\;M^{\prime }(0)=0,\;N(0)=1,\\ F\left(\infty \right)=0,\;G(\infty )=0,\;H\left(0\right)=0,\;\Theta \left(\infty \right)=0,\;f\left(\infty \right)=0,\;M\prime (\infty )=0,\;N(\infty )=0. \end{array}$$
where Pr=ν/α is the Prandtl number, $Sc=v/D_{B}$ is the Schmidt number, $\beta_{1}=\lambda_{1}c$ the Deborah number, while suction/injection parameter, Brownian motion, variable viscosity, thermophoresis parameter, magnetic parameter, temperature ratio, and thermal radiation are defined as [31]: (25)$$Ws=\frac{w}{\sqrt{c\nu }},Nb=\frac{\tau^{*}D_{B}\left(C_{w}-C_{\infty }\right)}{v},\delta =\zeta \left(\tau_{w}-\tau_{\infty }\right),Nt=\frac{\tau^{*}D_{\tau }\left(\tau_{w}-\tau_{\infty }\right)}{\tau_{\infty }v},M=\frac{\sigma B_{0}^{2}}{c_{p}},Rd=\frac{4\sigma^{*}T_{\infty }^{3}}{kk^{*}}.$$

The skin friction $Cf_{x}$, local Sherwood $Sh_{r}$ and Nusselt number $Nu_{r}$ are mathematically can be written as [32,33]:(26)$$Cf=\frac{\sqrt{\tau_{zr}^{2}+}\tau_{z\varphi }^{2}}{\rho {\left(\Omega r\right)}^{2}},$$
(27)$$Sh_{r}=-\frac{r}{\left(C_{w}-C_{\infty }\right)}\left(\frac{\partial \tau }{\partial C}\right){|}_{z=0},$$
(28)$$Nu_{r}=\frac{r}{\left(\tau_{w}-\tau_{\infty }\right)}\left[1+\frac{16\sigma^{*}\tau^{3}}{3kk^{*}}\right]\left(\frac{\partial \tau }{\partial z}\right){|}_{z=0}.$$

The skin friction, Sherwood and Nusselt numbers have a non-dimensional structure as:(29)CfRer1/2=f″2(0)+g′2(0),
(30)$$Re^{-}\frac{1}{2}Sh_{r}=-\phi \left(0\right),$$
(31)$$Re^{-}\frac{1}{2}Nu_{r}=-(1+\frac{4}{3}Rd{\left\{1+\left(\Theta_{w}-1\right)\Theta \left(0\right)\right\}}^{3})\Theta^{\prime }\left(0\right).$$

Here $Re=\frac{ru}{\nu }$ is the local Reynold number. 

## 4. Solution Procedures

The higher-order model equation is brought down to first order by choosing variables: (32)$$\begin{array}{l} \chi_{1}=H,\;\chi_{2}=F,\;\chi_{3}=F\prime ,\;\chi_{4}=G,\;\chi_{5}=G\prime ,\;\chi_{6}=\theta ,\;\chi_{7}=\theta \prime ,\;\\ \chi_{8}=\phi ,\;\chi_{9}=\phi \prime ,\;\chi_{10}=M,\;\chi_{11}=M\prime ,\;\chi_{13}=N,\;\chi_{14}=N\prime . \end{array}\}$$


$\chi_{1}^{\prime }=-2\chi_{2},\chi_{1}^{\prime }=\chi_{3},$



$\begin{array}{l} \chi_{3}^{\prime }=\chi_{7}\chi_{3}+\frac{e^{\delta \theta }}{\delta }\{S(\frac{\eta \chi_{3}}{2}+\chi_{2})+\chi_{2}^{2}+\chi_{1}\chi_{2}^{2}-\chi_{4}^{2}\}+\beta_{1}\frac{e^{\delta \theta }}{\delta }\{\chi_{1}\chi_{3}+2\chi_{1}\chi_{2}\chi_{3}-2\chi_{1}\chi_{4}\chi_{5}\}\\ M_{0}\frac{e^{\delta \theta }}{\delta }(\chi_{2}+\beta_{1}\chi_{1}\chi_{3}),\chi_{4}^{\prime }=\chi_{5}, \end{array}$



$\chi_{5}^{\prime }=\frac{\delta \chi_{5}\chi_{7}+e^{\delta \theta }\{S(\frac{\chi_{5}-\eta }{2})+2\chi_{2}\chi_{4}+\chi_{1}\chi_{5}\}+\beta_{1}\{2\chi_{1}\chi_{2}\chi_{5}-2\chi_{1}\chi_{4}\chi_{2}\}+M_{0}e^{\delta \theta }(\chi_{4}+\beta_{1}\chi_{1}\chi_{5})}{1+\beta_{1}e^{\delta \theta }\chi_{1}^{2}},$



$\begin{array}{l} \chi_{6}^{\prime }=\chi_{7},\chi_{7}^{\prime }=\frac{\frac{4}{3}Rd\chi_{7}^{2}(\theta_{w}-1)\{3\chi_{6}^{2}{\left(\theta_{w}-1\right)}^{2}+6\chi_{6}(\theta_{w}-1)-3\}-\mathrm{Pr}\{Nb\chi_{7}\chi_{9}+Nt\chi_{7}^{2}-A\chi_{7}\chi_{10}-\chi_{1}\chi_{7}\}}{1-\frac{4}{3}Rd-\frac{4}{3}Rd\chi_{6}(\theta_{w}-1)\{(\theta_{w}-1)\chi_{6}^{2}-3\chi_{6}(\theta_{w}-1)+3\}},\\ \chi_{8}^{\prime }=\chi_{9}, \end{array}$



$\begin{array}{l} \chi_{9}^{\prime }=\frac{Nt}{Nb}\left(\frac{\frac{4}{3}Rd\chi_{7}^{2}(\theta_{w}-1)\{3\chi_{6}^{2}{\left(\theta_{w}-1\right)}^{2}+6\chi_{6}(\theta_{w}-1)-3\}-\mathrm{Pr}\{Nb\chi_{7}\chi_{9}+Nt\chi_{7}^{2}-A\chi_{7}\chi_{10}-\chi_{1}\chi_{7}\}}{1-\frac{4}{3}Rd-\frac{4}{3}Rd\chi_{6}(\theta_{w}-1)\{(\theta_{w}-1)\chi_{6}^{2}-3\chi_{6}(\theta_{w}-1)+3\}}\right)\\ Sc(A\chi_{9}\chi_{10}+\chi_{91}\chi_{9}), \end{array}$
(33)$$\begin{array}{l} \chi_{10}^{\prime }=\chi_{11},\chi_{11}^{\prime }=Bt[-2\chi_{11}\chi_{2}-\chi_{1}\chi_{11}+\chi_{2}\chi_{14}+\chi_{13}\chi_{2}+S(\frac{\eta \chi_{12}}{2}+\chi_{11})],\\ \chi_{12}^{\prime }=\chi_{13},\chi_{13}^{\prime }=Bt[2\chi_{11}\chi_{1}+2\chi_{13}\chi_{2}+\frac{S}{2}(\eta \chi_{14}+\chi_{13})] \end{array}$$


The boundary conditions are:(34)$$\begin{array}{l} \chi_{1}(0)=1,\chi_{2}(0)=0,\chi_{4}(0)=1,\chi_{6}(0)=1,\chi_{8}(0)=1,\chi_{11}(0)=0,\chi_{13}(0)=1,\\ \chi_{2}(\infty )=0,\chi_{4}(\infty )=0,\chi_{6}(\infty )=0,\chi_{8}(\infty )=0,\chi_{11}(\infty )=1,\chi_{13}(\infty )=0. \end{array}$$

## 5. Results and Discussion

The discussion section is devoted to understanding better the graphical and physical description. The system of non-linear Equations (18)–(23) along with their boundary conditions, Equation (24), are solved through numerical method bvp4c. The configuration of the problem is described in Figure 1. The velocities, energy profile, concentration distribution ϕ(η), magnetic strength in the radial direction M(η), and azimuthal magnetic strength N(η) are explored graphically through different physical constraints Figure 2, Figure 3, Figure 4, Figure 5, Figure 6, Figure 7, Figure 8, Figure 9 and Figure 10. While keeping Pr=6.7, ω=1.0, δ=0.5, β=0.1, M=1.1, Nb=0.5, Nt=0.7, $Q_{w}=1.2$, Sc=2.0, and Rd=0.5. 

Figure 2a–c depicts the behavior velocities profiles against the variation of Deborah number β. All three velocity shows decreasing behavior for incremented of β. Deborah number is the measure of the content evaluation period to content recreational time, so having optimum stress relaxation or eliminating observation time increases the value of β. It reflects the fluid’s solid-like reaction. The hydrodynamic boundary layer thins out, and the velocity experiences more resistance.

The implications of the suction factor (Ws<0) on the velocities profile and heat spectrum on the disc surface are depicted in Figure 3a–c. It seems to be that as the suction velocity rises, the velocity decreases. Because the suction velocity draws the fluid particles towards the pores in the disk, which causes the momentum boundary layer reduction. The enhancement of suction velocity also decreases the fluid temperature.

Figure 4a–d are drawn to depict the effects of injection velocity parameter (Ws>0) on the radial, azimuthal, tangential velocity profile and temperature distribution. It can be seen that the enhancing of injection velocity increases the velocity and fluid temperature.

The radial, azimuthal, tangential velocity profile and temperature distribution variation are illustrated in Figure 5a–d. Figure 5a depicts the dominant behavior of radial velocity with ω. Physically, fluid particles are moved in a radial direction owing to centrifugal force with the enhancement of parameters ω. The increasing value of ω shows that the rotation parameter becomes greater than extending. The radial velocity exceeds the disc stretching velocity too close to the disc stretching surface. It does, however, gradually fade away from the disc. It is concluded that the impact of centrifugal force is limited and dominant in the vicinity of the disk’s surface. Besides, angular velocity G(η) near the disk flourished shown in Figure 5b. The axial velocity increasing against the strengthening of ω is illustrated in Figure 5c. Figure 5d demonstrates the decreases of temperature θ(η) with ω. Physically, a faster spinning disc reduces the width of the thermal boundary, which play a significant role in the cooling of the system.

Figure 6a–d are rough sketches that show the effects of the viscosity factor on the velocity and temperature spectrum on the disk’s surface. The viscosity parameter causes radial velocity, tangential velocity, and temperature profile to increase, while azimuthal velocity reduces.

Figure 7a,b indicates how the heat transfer changes when the Prandtl number *Pr* and the thermal radiation factor change. The thermal dispersion of a strong Prandtl fluid is low, while the thermal diffusivity of a low Prandtl fluid is high. Figure 7b is drawn to explore the influence of radiation parameters Rd on the thermal mechanism of fluid. With a higher value of the radiation parameter, an increasing trend in heat is analyzed. Because, with enhancing of thermal radiation the fluid absorbs more heat, and as a result increment occurs in boundary layer thickness and fluid temperature. Figure 7c is sketched to illustrate temperature θ(η) variation versus temperature ratio $Q_{w}$. While Figure 7d indicates the consequence of molecular diffusion on the thermal performance of a Maxwell nanofluid. The Brownian motion produces random movement between fluid particles, which generate more heat, and as a result, the fluid temperature increases.

Figure 8a,b accordingly represents the action of the concentration field as a function of the thermophoresis term and Brownian. It can be shown that with *Nt*, the nanofluid concentration field raises while with *Nb*, the concentration profile drops. The suction parameter (Ws)<0 and Schmidt number *Sc* effect are illustrated in Figure 8c and Figure 8d respectively. The concentration field reduces with an increase of both suction velocity and Schmidt number. The consequences of different parameters (Bt, Ws) on the axial and radial magnetic strength profiles are illustrated through Figure 9a–d respectively. It can observe that both parameters Batchlor number *Bt* and injection parameter *Ws* positively effect the magnetic strength profile along axial and radial direction.

Figure 10a,b express the nature of heat $-\theta^{\prime }\left(0\right)$ and mass transfer $-\phi^{\prime }\left(0\right)$ against radiation parameter *Rd* and Brownian motion parameter *Nb*, respectively. Because of the improving effect of radiation, the fluid temperature also rises, which enhances the heat transmission rate. Table 1 displays the numerical outcomes for skin friction and compared it with the published literature. Table 2 illustrates the comparison of bvp4c and RK4 techniques for the numerical outcomes. The numerical outputs for Sherwood number $Sh_{r}$ and Nusselt number $Nu_{r}$ are plotted in Table 3.

## 6. Conclusions

In the present mathematical model, the Maxwell nanoliquid flow over a porous spinning disk with suction/injection effects has been examined. The flow is studied in the context of magnetization and radiation. The numerical results are found through bvp4c, while for comparison purposes, the computation is carried out via the RK4 technique. From the above studies the following conclusions have been drawn:

The centrifugal force is more effective and dominant near the disk surface, so causes the maximum velocity in the neighborhood of the surface of the disk.The fluid velocity reduces by the action of suction effects. Because the suction effect gripping the fluid particles towards the pores of the disk, which causes the momentum boundary layer reduction.The axial, radial and tangential velocities are boosts with the increases of injection parameter. The enhancement in rotation parameter also boosts the azimuthal and radial flows. The thermal energy profile enhances by the consequence of Brownian motion parameter *Nb*. The Brownian motion produces random movement between fluid particles, which generate more heat, as a result, the fluid temperature increases.The fluid temperature decline with Prandtl number Pr, while incline with thermal radiation parameter, Rd. The nanofluid concentration field enhances with Nt, while Nb causes the reduction of concentration profile.

## Figures and Tables

**Figure 1 micromachines-12-00540-f001:**
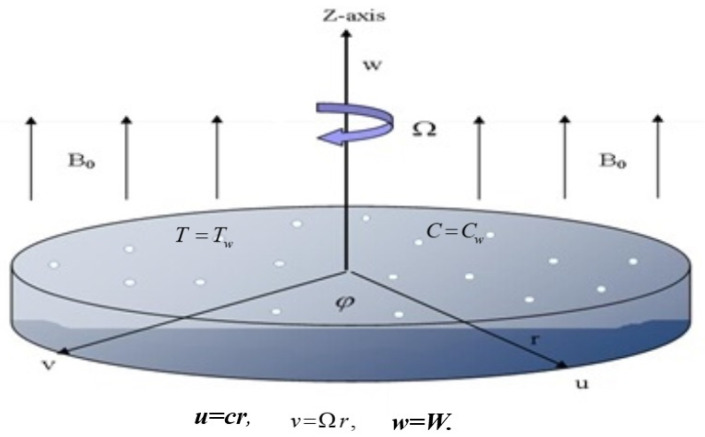
Stretchable porous rotating disk.

**Figure 2 micromachines-12-00540-f002:**
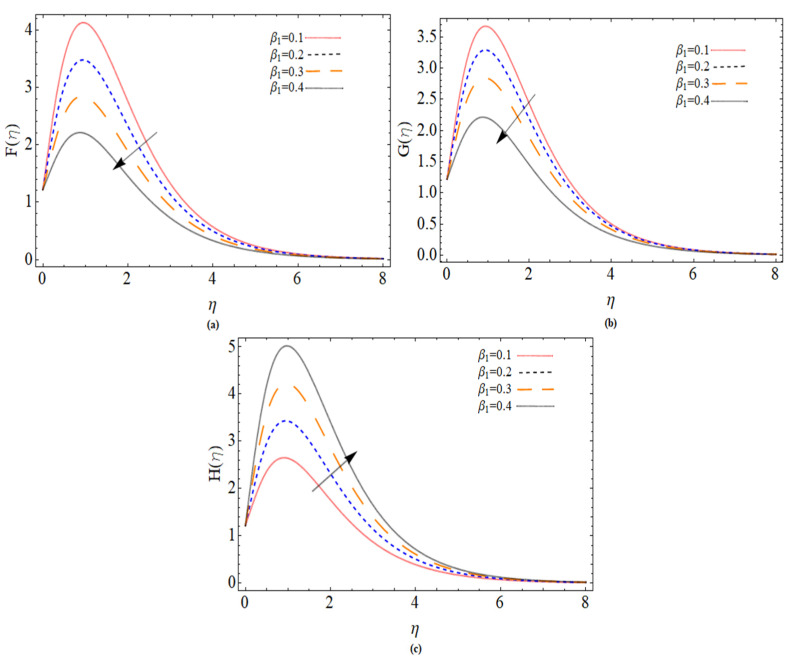
The out-turn of Deborah number β on the axial, radial and azimuthal velocity profiles, respectively.

**Figure 3 micromachines-12-00540-f003:**
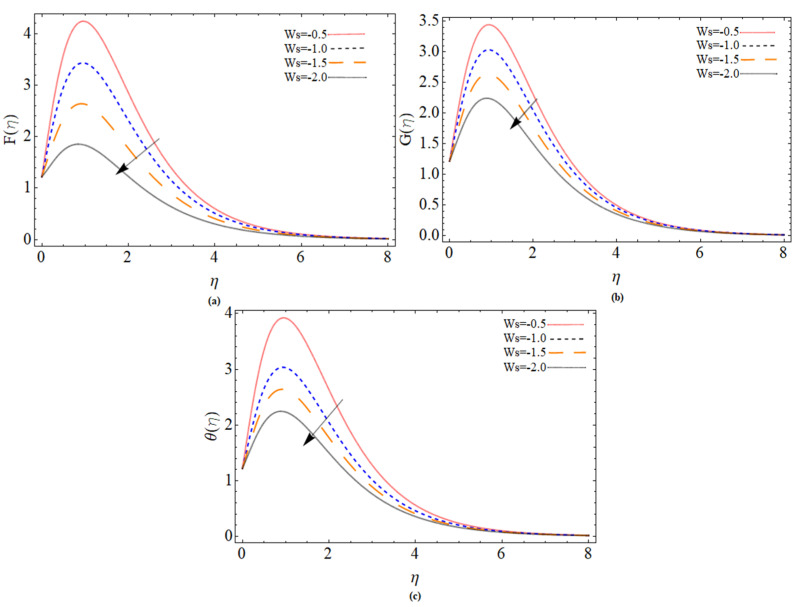
The out-turn of the suction parameter (Ws<0) on the axial velocity, radial velocity and temperature distribution profiles, respectively.

**Figure 4 micromachines-12-00540-f004:**
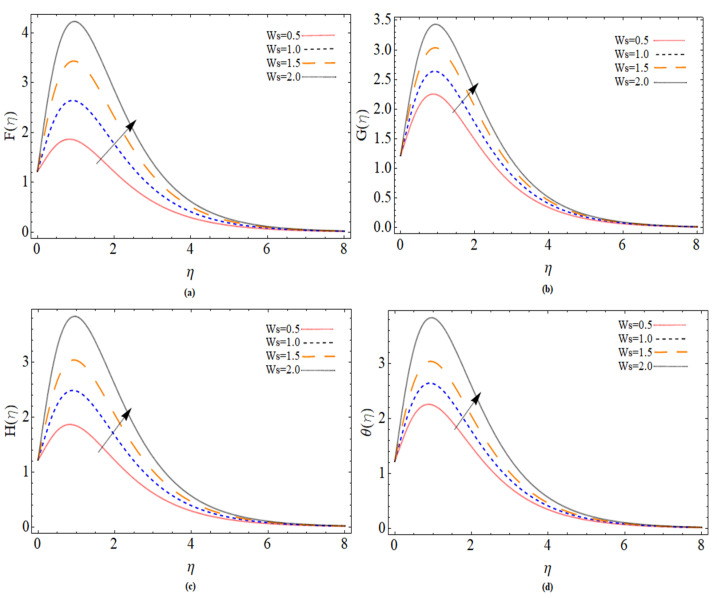
The out-turn of injection parameter (Ws>0) on the axial, radial and azimuthal velocity profiles and temperature distribution, respectively.

**Figure 5 micromachines-12-00540-f005:**
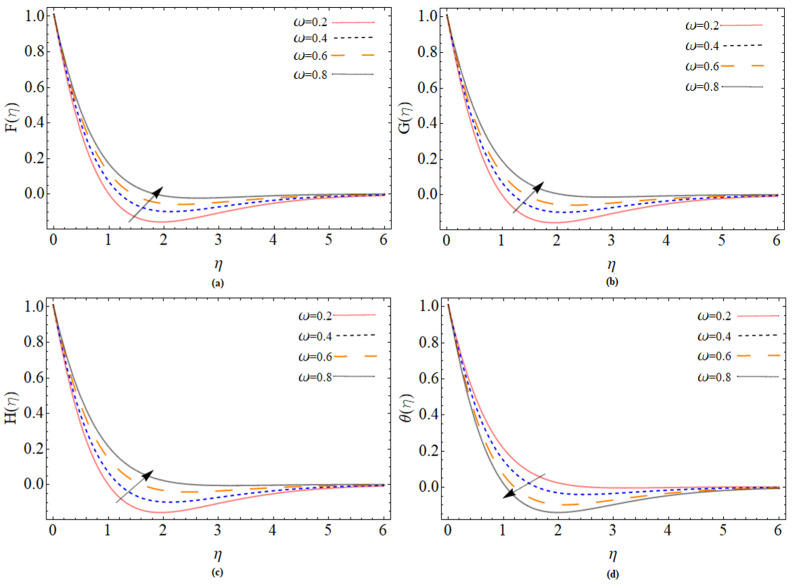
The out-turn of rotation parameter ω on the axial, radial and azimuthal velocity and temperature profiles, respectively.

**Figure 6 micromachines-12-00540-f006:**
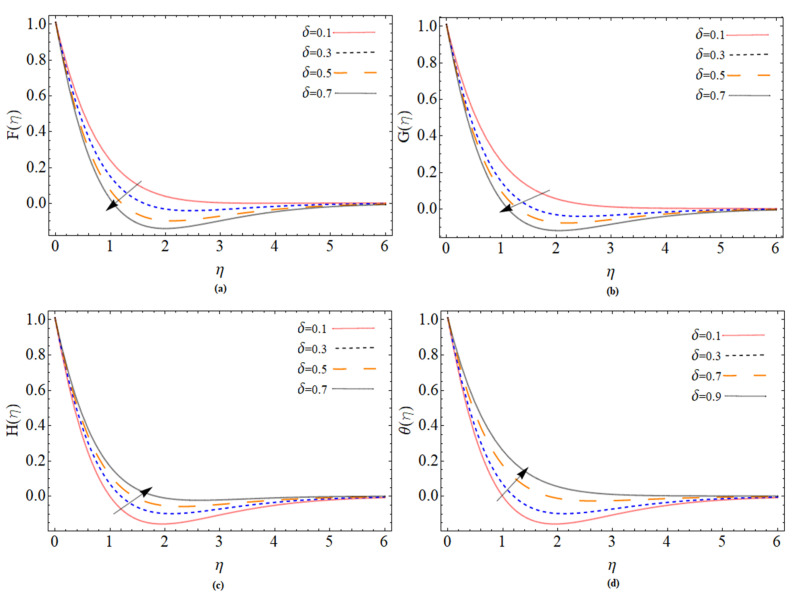
The out-turn of viscosity parameter δ on the axial, radial and azimuthal velocity and temperature profiles, respectively.

**Figure 7 micromachines-12-00540-f007:**
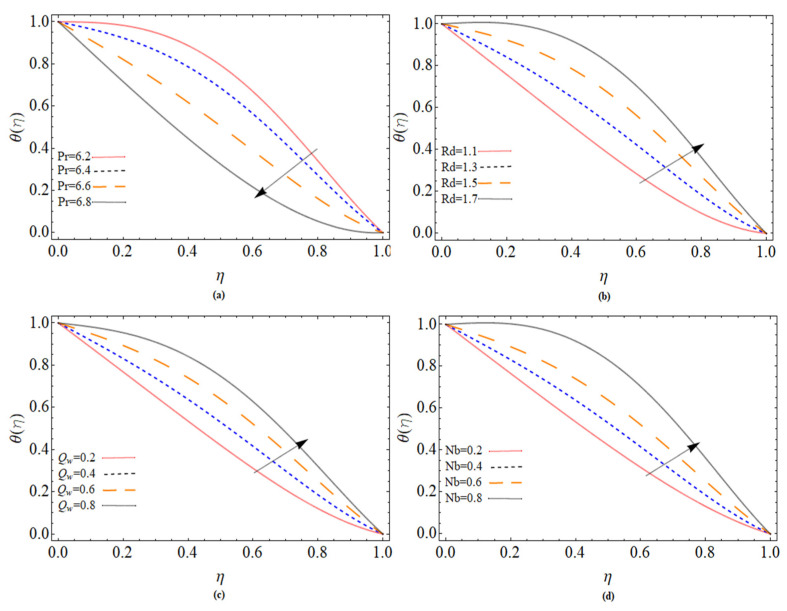
The out-turn of different parameters (Pr, Rd, Θ, Nb ) on the temperature distribution profile.

**Figure 8 micromachines-12-00540-f008:**
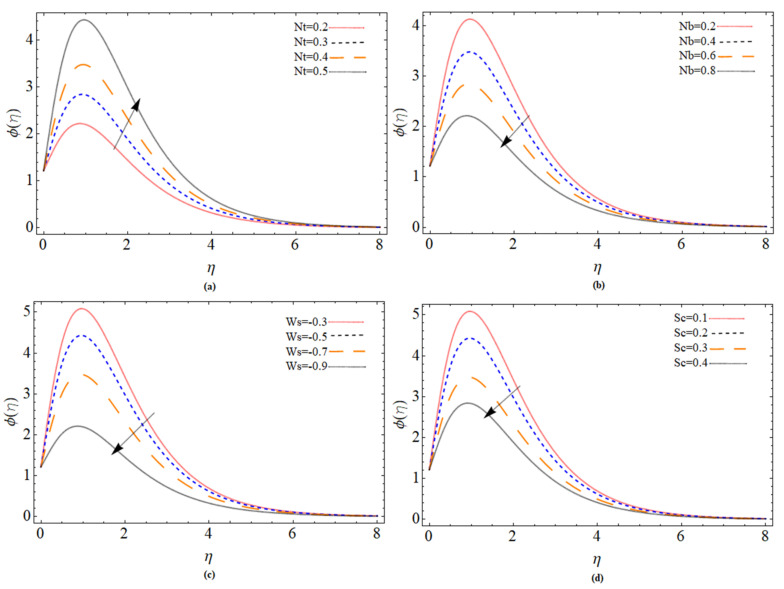
The out-turn of different parameters (Nt, Nb, Ws, Sc ) on the concentration profile.

**Figure 9 micromachines-12-00540-f009:**
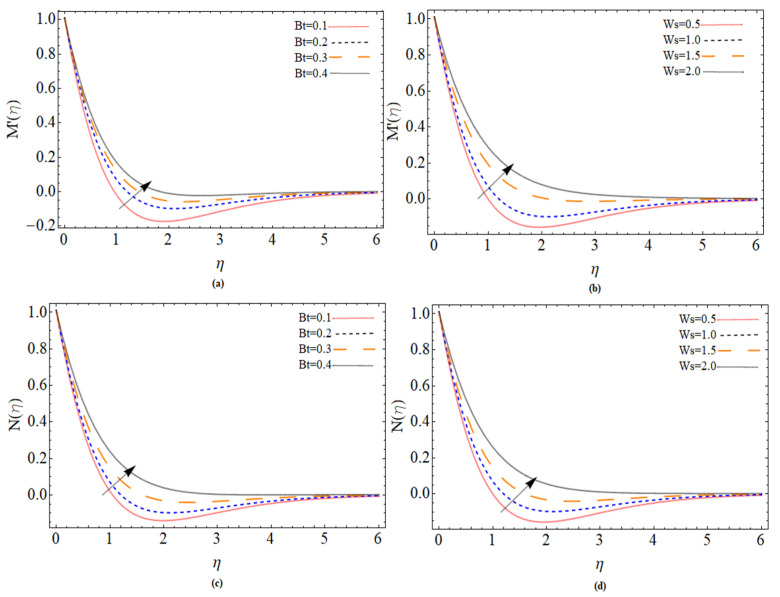
The out-turn of different parameters (Bt, Ws ) on the axial and radial magnetic strength profiles, respectively.

**Figure 10 micromachines-12-00540-f010:**
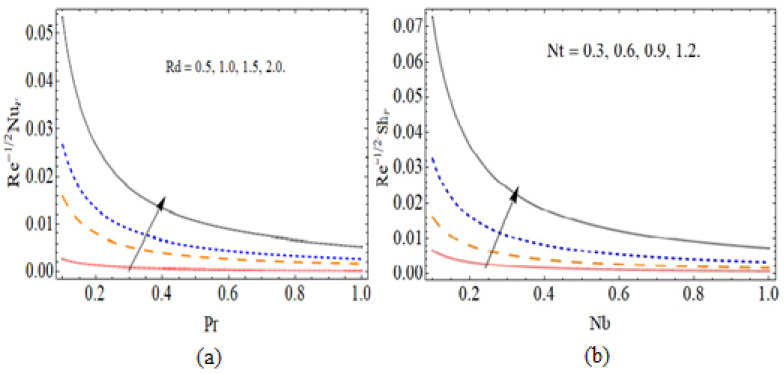
The out-turn of radiation parameter and Brownian coefficient versus the heat and mass transmission profiles, respectively.

**Table 1 micromachines-12-00540-t001:** The numerical outcomes for skin friction $F^{\prime }\left(0\right)$.

ω	Mustafa et al. [7]	Ahmed et al. [31]	Present Paper
0	−1.1737	−1.1379	−1.1380
1	−0.9483	−0.9485	−0.9487
2	−0.3262	−0.3264	−0.3266
5	3.1937	3.11937	3.11939
10	12.7209	12.7209	12.7811
20	40.9057	40.9057	40.9058

**Table 2 micromachines-12-00540-t002:** Comparison between Runge Kutta order four and bvp4c method.

η	RK4	bvp4c	Absolute Error
1.0	1.000000	1.000000	8.146310$\times 10^{-13}$
1.2	1.199831	1.199831	3.387821 $\times 10^{-9}$
1.4	0.988459	0.988459	2.845561 $\times 10^{-9}$
1.6	0.879189	0.879189	2.813281 $\times 10^{-9}$
1.8	0.539393	0.539393	3.287961 $\times 10^{-9}$

**Table 3 micromachines-12-00540-t003:** The comparison of RK4 and Bvp4c for Sherwood $Sh_{r}$ and Nusselt number $Nu_{r}$, while keeping ω=1.0, δ=0.5, β=0.1, M=1.1, Nb=0.5, Nt=0.7, $Q_{w}=1.2$ and Rd=0.5.

$Sh_{r}$			$Nu_{r}$		
Pr	Bvp4c	RK4	Sc	Bvp4c	RK4
3.0	0.7718284	0.7718285	1.0	1.457986	1.457995
4.0	0.6999885	0.6999885	1.5	1.531211	1.531220
5.0	0.6290089	0.6290088	2.0	1.596949	1.596949
6.0	0.5632372	0.5632370	2.5	1.685639	1.685639

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
