# Peer review of "Numerical Analysis of Thermal Radiative Maxwell Nanofluid Flow Over-Stretching Porous Rotating Disk"

_micromachines, 2021, doi:10.3390/mi12050540_

Round 1

Reviewer 1 Report

This paper is about Maxwell flow over-stretching porous rotating disk.

The study is based on simple PDEs rather than Navier-Stokes equations.

So, the reviewer has some doubts about the accuracy of the present study.

Before it is resolved, the reviewer cannot agree with its publication.

Author Response

REPLY TO REVIEWER. 1

Comment 1:  This paper is about Maxwell flow over-stretching porous rotating disk. The study is based on simple PDEs rather than Navier-Stokes equations. So, the reviewer has some doubts about the accuracy of the present study. Before it is resolved, the reviewer cannot agree with its publication.

Author response: Thank you so much. We appreciate your consideration, guidance, help and time. In the present manuscript, we have numerically solved the system of highly nonlinear partial differential equations consist of Navier-stokes equations along with Maxwell equations (See Eqs. 1-13). For the validity and accuracy of the results, the outcomes are compared through Table 1-3.

Reviewer 2 Report

In recent times heat transfer technology is confronted with increasing demand of cooling applications of miniaturised high heat flux components. However, the fluids that are traditionally used for heat transfer applications such as water, oils and ethylene glycol have a rather low thermal conductivity and do not meet the growing demand as an efficient heat transfer agent. The conventional method for increasing heat dissipation is to increase the area available for exchanging heat with a heat transfer fluids, but this approach requires an undesirable increase in the size of thermal management system. Considering the rising demands of modern technology, including chemical production, power stations and microelectronics, there is a need to develop new types of fluids that will be more effective in terms of heat exchange performance. Nanofluids are solid–liquid composite materials consisting of solid nanoparticles or nanofibers, with sizes typically on the order of 1–100 nm, suspended in a liquid. And nanofluids technology has emerged as a new heat transfer technique in recent years. Applications to superconducting magnets and most importantly superfast computing are posing tremendous challenge to thermal management. Thermal properties of nanofluids have been a topic of intense research during the past few decades due to their perspective technological applications in electronics cooling and heat transfer. In this manuscript, the Maxwell nanoliquid flow over porous spinning disk with suction/injection effects has been examined. The flow is studied in the context of a magnetization and radiation. This manuscript should be rejected for published in Micromachines. However, if the authors are willing to make the substantial revisions according to my comments, I would be glad to re-review this manuscript. Here are my detailed comments:

  1. The authors need to reorganize the current introduction, which normally consists of three parts at least: background, literature review, brief of the proposed work. The current one is nothing but a literature review. Why their work is important comparing to previous reports? I think this is essential to keep the interest of the reader.
  2. In Figs. 2-6, the authors should give the explanations for the difference of data collected from different sources.
  3. Please check all Equations double.
  4. It is suggested to discuss what the main advantages the proposed mathematical model has.
  5. I am quite interested in some parametric study with the proposed mathematical model. The manuscript could be more substantial if the authors do so. At least, the authors need to write some statements that how the proposed mathematical model can be used for the parametric study.
  6. Although the results look “making sense”, the authors should dig deeper in the results by presenting some in-depth discussion, such as implications of the results, such as possible application of them.
  7. The centrifugal force is more effective and dominant near the disk surface, so causes the maximum velocity in neighborhood of surface of the disk. The mass and heat transfer rates is significantly reduced with Brownian diffusion. The authors should give some explanation on above conclusions.
  8. Nanofluids have been widely used in the industry. Nanofluid, characterized by a significant increase in molecular mobility compared to conventional engineered fluid [ASME, 1995], is found to serve in many practical applications, such as porous materials [Fractals, 2021, 29(1):2150017; International Journal of Heat and Mass Transfer, 2019, 137:365-371], and has been drawn particular attentions for encapsulation in medical industry in recent years as a result. Authors should introduce some related knowledge to readers. I think this is essential to keep the interest of the reader.
  9. Please, expand the conclusions in relation to the specific goals and the future work.

Author Response

REPLY TO REVIEWER. 2

In recent times heat transfer technology is confronted with increasing demand of cooling applications of miniaturised high heat flux components. However, the fluids that are traditionally used for heat transfer applications such as water, oils and ethylene glycol have a rather low thermal conductivity and do not meet the growing demand as an efficient heat transfer agent. The conventional method for increasing heat dissipation is to increase the area available for exchanging heat with a heat transfer fluids, but this approach requires an undesirable increase in the size of thermal management system. Considering the rising demands of modern technology, including chemical production, power stations and microelectronics, there is a need to develop new types of fluids that will be more effective in terms of heat exchange performance. Nanofluids are solid–liquid composite materials consisting of solid nanoparticles or nanofibers, with sizes typically on the order of 1–100 nm, suspended in a liquid. And nanofluids technology has emerged as a new heat transfer technique in recent years. Applications to superconducting magnets and most importantly superfast computing are posing tremendous challenge to thermal management. Thermal properties of nanofluids have been a topic of intense research during the past few decades due to their perspective technological applications in electronics cooling and heat transfer. In this manuscript, the Maxwell nanoliquid flow over porous spinning disk with suction/injection effects has been examined. The flow is studied in the context of a magnetization and radiation. This manuscript should be rejected for published in Micromachines. However, if the authors are willing to make the substantial revisions according to my comments, I would be glad to re-review this manuscript. Here are my detailed comments:

Comment 1: The authors need to reorganize the current introduction, which normally consists of three parts at least: background, literature review, brief of the proposed work. The current one is nothing but a literature review. Why their work is important comparing to previous reports? I think this is essential to keep the interest of the reader.

Author response: As instructed by the reviewer, we have arranged the introduction section consisting of background, literature review and brief of the proposed work. In the last paragraph of literature review, we have highlighted the novelty of the current work.

Comment 2: In Figs. 2-6, the authors should give the explanations for the difference of data collected from different sources.

Author response: In the revised manuscript, we have tried to explain more effectively the physical mechanism behind each figure.

Comment 3: Please check all Equations double.

Author response: Rectified.

Comment 4: It is suggested to discuss what the main advantages the proposed mathematical model has.

Author response: In the revised manuscript, we have discussed in detail the advantages of the proposed mathematical model.

Comment 5: I am quite interested in some parametric study with the proposed mathematical model. The manuscript could be more substantial if the authors do so. At least, the authors need to write some statements that how the proposed mathematical model can be used for the parametric study.

Author response: The authors are thankful for the reviewer comments. But we are unable to understand this comment.

Comment 6: Although the results look “making sense”, the authors should dig deeper in the results by presenting some in-depth discussion, such as implications of the results, such as possible application of them.

Author response: In the revised manuscript, we have tried to explain more effectively the results and physical mechanism behind each figure.

Comment 7: The centrifugal force is more effective and dominant near the disk surface, so causes the maximum velocity in neighborhood of surface of the disk. The mass and heat transfer rates is significantly reduced with Brownian diffusion. The authors should give some explanation on above conclusions.

Author response: Physically, fluid particles are moved in a radial direction owing to centrifugal force. The increasing value of  shows that rotation parameter become greater than extending. The radial velocity exceeds the disc stretching velocity too close to the disc stretching surface. It does, however, gradually fade away from the disc. It is concluded that the impact of centrifugal force is limited and dominant in the vicinity of the disk's surface as shown in Figure 5(a).

Rectified.

Comment 8: Nanofluids have been widely used in the industry. Nanofluid, characterized by a significant increase in molecular mobility compared to conventional engineered fluid [ASME, 1995], is found to serve in many practical applications, such as porous materials [Fractals, 2021, 29(1):2150017; International Journal of Heat and Mass Transfer, 2019, 137:365-371], and has been drawn particular attentions for encapsulation in medical industry in recent years as a result. Authors should introduce some related knowledge to readers. I think this is essential to keep the interest of the reader.

Author response: We have tried to introduce most related literature and knowledge in the revised manuscript.

Comment 9: Please, expand the conclusions in relation to the specific goals and the future work.

Author response: We have expand the conclusion section of the revised manuscript

Reviewer 3 Report

Report on Numerical analysis of thermal radiative Maxwell nanofluid flow over-stretching porous 3 rotating disk

General Overview:

In this research the authors proposed to deal with fluid flow over the rotating disk using various important physical parameters that are critically important in industries, engineering, and scientific implementation. The traditional swirling flow of Von Karman is optimized for Maxwell fluid over a porous spinning disc with a consistent suction/injection effect. Buongiorno’s model is involved to emphasis on the Brownian motion and thermophoresis. RK45 with shooting technique is involved to solve the system for final solutions. The work is interesting and quite innovative. However, there are some gaps in the study which must be addressed before publication.

Comments:

  1. The introduction part lacks in two major areas nanofluids and the Porous medium. Since, the disk is considered under Darcy model (Porous medium), therefore, there should be a note about Darcy medium to link the problem with literature. Recent articles such as: Numerical Scrutinization of Darcy-Forchheimer Relation in Convective Magnetohydrodynamic Nanofluid Flow Bounded by Nonlinear Stretching Surface in the Perspective of Heat and Mass Transfer. Micromachines2021, Numerical Exploration of the Features of Thermally Enhanced Chemically Reactive Radiative Powell-Eyring Nanofluid Flow via Darcy Medium over Non-linearly Stretching Surface Affected by a Transverse Magnetic Field and Convective Boundary Conditions,  Applied Nanoscience, Consequences of Soret-Dufour Effects, Thermal Radiation, and Binary Chemical Reaction on Darcy Forchheimer Flow of Nanofluids, 2020, Symmetry, Darcy-Forchheimer nanofluidic flow manifested with Cattaneo-Christov theory of heat and mass flux over non-linearly stretching surface, PLoS ONE, Significance of thermal slip and convective boundary conditions on three dimensional rotating Darcy-Forchheimer nanofluid flow, Symmetry, Finite Element Study of Magnetohydrodynamics (MHD) and Activation Energy in Darcy-Forchheimer Rotating Flow of  Casson Carreau Nanofluid, Processes, and cross references cited therein can be very handy in this regard.
  2. The diagram of geometry is lacking the boundary conditions to give a clear insight to the reader. Improve the diagram and put all the boundary conditions within the diagram.
  3. Check equations (15) and (17).
  4. The authors have considered Sherwood number and Nusselt number but they do not consider the skin-friction and no expression is provided, although it is very important in Darcy type of medium. Why?
  5. Discussion must be improved with more logical reasoning besides the increasing decreasing trends.
  6. The data and Graphs for Nusselt number And Sherwood numbers are missing.
  7. Add more results in abstract and conclusions.
  8. Improve the reference list and add more recently published articles especially those published within last two – three years in journal closely related with Micromachines.

Author Response

REPLY TO REVIEWER. 3

General Overview:

In this research the authors proposed to deal with fluid flow over the rotating disk using various important physical parameters that are critically important in industries, engineering, and scientific implementation. The traditional swirling flow of Von Karman is optimized for Maxwell fluid over a porous spinning disc with a consistent suction/injection effect. Buongiorno’s model is involved to emphasis on the Brownian motion and thermophoresis. RK45 with shooting technique is involved to solve the system for final solutions. The work is interesting and quite innovative. However, there are some gaps in the study which must be addressed before publication.

Comment 1: The introduction part lacks in two major areas nanofluids and the Porous medium. Since, the disk is considered under Darcy model (Porous medium), therefore, there should be a note about Darcy medium to link the problem with literature. Recent articles such as:

  • Numerical Scrutinization of Darcy-Forchheimer Relation in Convective Magnetohydrodynamic Nanofluid Flow Bounded by Nonlinear Stretching Surface in the Perspective of Heat and Mass Transfer. Micromachines2021, 
  • Numerical Exploration of the Features of Thermally Enhanced Chemically Reactive Radiative Powell-Eyring Nanofluid Flow via Darcy Medium over Non-linearly Stretching Surface Affected by a Transverse Magnetic Field and Convective Boundary Conditions,  Applied Nanoscience,
  • Consequences of Soret-Dufour Effects, Thermal Radiation, and Binary Chemical Reaction on Darcy Forchheimer Flow of Nanofluids, 2020, Symmetry,
  • Darcy-Forchheimer nanofluidic flow manifested with Cattaneo-Christov theory of heat and mass flux over non-linearly stretching surface, PLoS ONE,
  • Significance of thermal slip and convective boundary conditions on three dimensional rotating Darcy-Forchheimer nanofluid flow, Symmetry,
  • Finite Element Study of Magnetohydrodynamics (MHD) and Activation Energy in Darcy-Forchheimer Rotating Flow of  Casson Carreau Nanofluid, Processes, and cross references cited therein can be very handy in this regard.

Author response: The reviewer suggested papers are consist of great interest and of similar techniques, which used have strengthen the References section and the techniques more plausible, therefore we have cited all of them in the revised manuscript.

Comment 2: The diagram of geometry is lacking the boundary conditions to give a clear insight to the reader. Improve the diagram and put all the boundary conditions within the diagram.

Author response: Rectified.

Comment 3: Check equations (15) and (17).

Author response: Rectified.

Comment 4: The authors have considered Sherwood number and Nusselt number but they do not consider the skin-friction and no expression is provided, although it is very important in Darcy type of medium. Why?

Author response: Rectified.

Comment 5: Discussion must be improved with more logical reasoning besides the increasing decreasing trends.

Author response: In the revised manuscript, we have tried to explain more effectively the results and physical mechanism behind each figure.

Comment 6: The data and Graphs for Nusselt number And Sherwood numbers are missing.

Author response: We have added Table 1-3 and Figure 10(a)-(b) in the revised manuscript for Nusselt number and Sherwood numbers.

Comment 7: Add more results in abstract and conclusions.

Author response: We have modified the abstract and conclusions as instructed by the reviewer.

Comment 8: Improve the reference list and add more recently published articles especially those published within last two – three years in journal closely related with Micromachines.

Author response: We have added some related work to the present article in the revised manuscript, which are recently published in journal closely related with Micromachines.

Round 2

Reviewer 1 Report

To the reviewer, the revised manuscript is appropriately modified according the comments.

Reviewer 2 Report

It is ok.

Reviewer 3 Report

I can see that most of the suggested corrections are addressed. I accept it.